# Assessment of Biological Activity of 28-Homobrassinolide via a Multi-Level Comparative Analysis

**DOI:** 10.3390/ijms24119377

**Published:** 2023-05-27

**Authors:** Junpeng Huang, Biaodi Shen, Xiao Rao, Xuehua Cao, Jianjun Zhang, Linchuan Liu, Jianming Li, Juan Mao

**Affiliations:** 1State Key Laboratory for Conservation and Utilization of Subtropical Agro-Bioresources, South China Agricultural University, Guangzhou 510642, China; huangjunpeng@stu.scau.edu.cn (J.H.); shenbiaodi@stu.scau.edu.cn (B.S.); xiaorao@stu.scau.edu.cn (X.R.); caoxuehua@stu.scau.edu.cn (X.C.); zhangjj@scau.edu.cn (J.Z.); lcliu@scau.edu.cn (L.L.); 2Guangdong Key Laboratory for Innovative Development and Utilization of Forest Plant Germplasm, College of Forestry and Landscape Architecture, South China Agricultural University, Guangzhou 510642, China; 3Department of Molecular, Cellular, and Developmental Biology, University of Michigan, Ann Arbor, MI 48109, USA

**Keywords:** brassinosteroids, 28-homobrassinolide, 24-epibrassinolide, brassinolide, bioactivity, BR response

## Abstract

Brassinosteroids (BRs) play vital roles in the plant life cycle and synthetic BRs are widely used to increase crop yield and plant stress tolerance. Among them are 24*R*-methyl-epibrassinolide (24-EBL) and 24*S*-ethyl-28-homobrassinolide (28-HBL), which differ from brassinolide (BL, the most active BR) at the C-24 position. Although it is well known that 24-EBL is 10% active as BL, there is no consensus on the bioactivity of 28-HBL. A recent outpouring of research interest in 28-HBL on major crops accompanied with a surge of industrial-scale synthesis that produces mixtures of active (22*R*,23*R*)-28-HBL and inactive (22*S*,23*S*)-28HBL, demands a standardized assay system capable of analyzing different synthetic “28-HBL” products. In this study, the relative bioactivity of 28-HBL to BL and 24-EBL, including its capacity to induce the well-established BR responses at molecular, biochemical, and physiological levels, was systematically analyzed using the whole seedlings of the wild-type and BR-deficient mutant of *Arabidopsis thaliana*. These multi-level bioassays consistently showed that 28-HBL exhibits a much stronger bioactivity than 24-EBL and is almost as active as BL in rescuing the short hypocotyl phenotype of the dark-grown *det2* mutant. These results are consistent with the previously established structure–activity relationship of BRs, proving that this multi-level whole seedling bioassay system could be used to analyze different batches of industrially produced 28-HBL or other BL analogs to ensure the full potential of BRs in modern agriculture.

## 1. Introduction

The improvement of important agronomic traits remains to be a major challenge for crop productivity in modern agriculture to meet the enormous demands of the rapid increasing human population. Brassinosteroids (BRs), a crucial plant growth-promotion hormone discovered in 1970 [1], were thought to have great potential to play essential roles in green agriculture [2], due to their roles in regulating important agronomic traits and modulating plant responses to a wide range of environmental stresses. One of the effective approaches to increasing crop yield is to cultivate new high-yielding or stress-tolerant plants through genetic engineering of genes involved in BR biosynthesis or signal transduction [3,4]. The direct exogenous application of BRs represents another ideal and technically simple approach to improving crop yield or stress resistance in modern agriculture. Their ability to function at an incredibly low concentration while being friendly to the environment are two important and appealing aspects of BRs. Among about 70 naturally occurring BRs, 24*R*-epibrassinolide (24-EBL, an epi-isomer of brassinolide) and 24*S*-28-homobrassinolide (28-HBL, carrying a 24*S* ethyl group) are widely used to study the physiological effects of steroid plant hormones on cereal crops, vegetables, and fruits [5]. It was reported that treatment with 28-HBL increased the yields of rice (*Oryza sativa*), wheat (*Triticum aestivum*), peanut (*Arachis hypogea*), mustard (*Brassica juncea*), potato (*Solanum tuberosum*), and cotton (*Gossypium hirsutum*) in a field experiment in India [6]. The study of Zhang et al. also showed that exogenous 24-EBL/28-HBL enhanced spikelet development in rice, thereby increasing the grain yield [7]. In a tomato study, 24-EBL/28-HBL significantly accelerated fruit ripening and improved their quality [8,9], such as enhancing starch levels through upregulating the expression of starch synthesis genes [10]. BRs can not only increase plant yield under favorable conditions, but also stimulate plant growth under challenging environmental conditions. 24-EBL/28-HBL increased tolerance to cold in maize [11] and cucumber (*Cucumis sativus*) [12], to heat in mustard (*Brassica juncea*) [13], to frost in winter wheat [14], to drought in maize (*Zea mays*) [15,16], to cadmium stress in tomato (*Solanum lycopersicum*) [17], and to lead stress in mustard (*Brassica juncea*) [18]. Exogenous application of BRs can also confer tolerance to plants to a wide spectrum of pathogen infections by inducing innate immune responses and activating several vital antioxidant enzymes [5,19], such as increasing the tolerance to postharvest disease in citrus fruit (*Citrus unshiu*) [20] and to fusarium diseases in barley (*Hordeum vulgare*) [21]. Thus, BRs can act as plant growth enhancers under favorable or unfavorable conditions when they are applied at the appropriate dose and at the correct stage of plant development [22]. In scientific studies, the extensive use of brassinolide (BL, the most active natural BR)/24-EBL in the laboratory greatly contributed to the elucidation of the BR biosynthetic pathway and the BR signaling cascade in Arabidopsis (*Arabidopsis thaliana*) [23,24,25,26,27,28,29,30] and the identification of several key BR biosynthetic enzymes and BR signaling components in major crop species, including rice (*Oryza sativa*) and maize (*Zea mays*) [5,31,32]. Therefore, both academic and industrial laboratories conduct intensive research on this fascinating plant hormone.

However, natural BRs are present at extremely low levels in plant tissues (0.01–0.1 ng/g fresh weight [FW] in leaves and shoots and 1–100 ng/g FW in pollen and seeds) [33]. BL (4 mg), the first BR whose structure was determined, was purified from roughly 40 kg of rapeseed (*Brassica napus* ssp. *napus*) pollen [34]. The extremely low levels of BRs in plants indicate that chemical synthesis is vital and becomes the only source of BRs for physiological studies in the laboratory and for practical applications in agriculture [35]. However, the chemical synthesis of BL is a rather costly endeavor with low yields due to the scarcity of synthetic precursors [35], thus limiting its use to laboratory experiments. By contrast, the chemical synthesis of 24-EBL and 28-HBL, which are also natural occurrences with low abundance in plants [36], is relatively cheap and could be scaled up for industrial production. This is largely due to readily available starting materials, ergosterol for 24-EBL [37] and stigmasterol for 28-HBL [38], and relatively efficient synthetic schemes [35,36]. Additionally, 24-EBL and 28-HBL show higher stability compared to BL in the field [22]. For these reasons, 24-EBL and 28-HBL are preferable to BL for large-scale agricultural applications.

Several highly sensitive techniques have been used to detect the biological activities of BRs and their analogs in an effort to understand the structure–activity relationship of this plant hormone and optimize its biological performance for applications in agriculture [35], such as the rice lamina inclination test (RLIT) [39], the bean (*Phaseolus vulgaris*) second internode test [34], and the soybean (*Glycine max*) epicotyl elongation assay [40], all relied on using explant segments. These studies identified key structural features essential for high biological activity of BRs, including the 24*S*-methyl/ethyl group, the 22*R*, 23*R*-vicinal diol on the side chain, the 2α, 3α-vicinal diol on the A-ring, 6-oxo/7-oxalactone functionality in the B-ring, and the trans-fused A/B ring junction [41,42]. These assays also demonstrated that BL with the 24*S*-methyl group is the most active member of the BR family, and the activity of 24-EBL is about 10% that of BL. By contrast, no clear consensus exists for the relative activity of 28-HBL. The earliest RLIT revealed similar activities for BL and 28-HBL [43], while the soybean epicotyl elongation assay indicated that the activity of 28-HBL is about 10% of BL [40]. The bean second internode assay, which aided the initial discovery of BR [34], showed that 28-HBL is much less active than 24-EBL [44]. It is important to note that the Japanese group, which claimed similar bioactivity of 28-HBL and BL, discovered that 28-HBL was only 10% and 1% active as BL in the radish and tomato bioassay, respectively [43]. The uncertainty about the bioactivity of 28-HBL might explain why simple searches at NCBI/PubMed returned 1068 results for 24-EBL but only 70 results for 28-HBL. To make matters worse is the presence of several stereoisomers of 28-HBL, including (22*R*,23*R*,24*S*)-28HBL (the most active one), (22*S*,23*S*,24*S*)-28-HBL, (22*R*,23*R*,24*R*)-28-HBL, and (22*S*,23*S*,24*R*)-28HBL with the first isomer being the most active and the other three only 1–3% active relative to BL [43]. It is well known that choices of chiral ligands/catalysts and purification schemes could easily lead to variability in the ratio of 22*S*,23*S*,*24S*-28HBL and 22*R*,23*R*,*24S*-28HBL in the final synthetic products derived from stigmasterol, consequently affecting the quality of industrially produced “28-HBLs” [42]. Together, these created a huge confusion in the scientific literature and commercial sources of 28-HBL, which can be easily revealed by a simple Google search with “22S,23S-28-homobrassinolide” or “22R,23R-28-homobrassinolide”. To ensure the full potential of 28-HBL and other synthetic BRs in modern green agriculture, a standard bioassay system using a well-characterized and easily available plant species is urgently needed to analyze the bioactivity of various “28-HBL” products and other commercially available synthetic BRs and/or BR analogs.

In the current study, the whole seedlings of the wild-type *Arabidopsis thaliana*, a widely used model system for plant biology studies, along with its BR-deficient dwarf mutant strain known as *de-etiolated 2* (*det2*) [23], were used to carry out a variety of bioassays at the molecular, biochemical, and physiological levels to analyze the bioactivities of 28-HBL. This study facilitates a better understanding of the biological activity of 28-HBL and provides a feasible multi-level bioassay system to evaluate the bioactivity of newly synthesized BRs and other potential BR analogs.

## 2. Results

### 2.1. 28-HBL Regulates BR-Responsive Gene Expression

It is well known that BR treatment feedback inhibits the expression of many BR biosynthetic genes, including *DWF4* (*DWARF4*) and *CPD* (*CONSTITUTIVE PHOTOMORPHOGENIC DWARF*), encoding two highly similar cytochrome P450 enzymes [45,46]. To compare the physiological activities of 28-HBL with BL and 24-EBL, we treated the 10-day-old wild-type Arabidopsis seedlings with the three BRs at two different concentrations and analyzed the transcript abundance of *DWF4* and *CPD* by quantitative real-time reverse transcription PCR (RT-qPCR). Our RT-qPCR analysis also included *SAUR-AC1* (*SMALL AUXIN UP RNA 1 FROM ARABIDOPSIS THALIANA ECOTYPE COLUMBIA*), a well-studied auxin-responsive gene that is also rapidly induced by BR [47]. As shown in Figure 1a, at the concentration of 1 μM, all three tested BRs resulted in great reduction in *DWF4* and *CPD* and marked induction of the *SAUR-AC1* gene. By contrast, at the concentration of 0.1 μM, the three BRs exhibited differential impacts on the expression of three tested BR-responsive genes. BL displayed the strongest activity, 24-EBL had the lowest activity, and 28-HBL exhibited an intermediate activity in reducing the *DWF4*/*CPD* transcript and inducing the expression of the *SAUR-AC1* gene (Figure 1b). Thus, lowering the concentrations of synthetic BRs can reveal their different impacts on the transcripts of known BR-responsive genes.

### 2.2. 28-HBL Efficiently Induces BES1 Dephosphorylation

BRs binding to their receptor BRI1 (Brassinosteroid-Insensitive1) triggers a well-defined protein phosphorylation cascade, leading to nuclear accumulation of un/de-phosphorylated forms of BES1 (BRI1-EMS-SUPPRESSOR 1) and BZR1 (BRASSINAZOLE-RESISTANT 1) [48], the two master transcription factors in the BR signaling pathway that change the expression patterns of thousands of BR-responsive genes [49]. Hence, BR-induced dephosphorylation of BES1 is widely used as a biochemical indicator of activated BR signaling. We treated the wild-type Arabidopsis seedlings with the three tested BRs at different concentrations, extracted total proteins, separated these total protein extracts by sodium dodecyl sulfate (SDS)-polyacrylamide gel electrophoresis (PAGE), and analyzed the BES1 phosphorylation status by an immunoblot with an anti-BES1 antibody. As indicated in Figure 2a, BES1 was present mainly as the slower-moving phosphorylated band in the untreated seedlings, and BL treatment resulted in disappearance of the phosphorylated BES1 band and accumulation of the faster-moving un/de-phosphorylated BES1 band in a concentration-dependent manner. Treatment of the wild-type seedlings with 28-HBL and 24-EBL resulted in similar changes in the BES1 phosphorylation status (Figure 2a). Interestingly, the un/de-phosphorylated BES1 accumulated more in seedlings treated with 0.5 μM BL than those treated with 28-HBL or 24-EBL. We repeated the experiments and quantified the ratios of un/de-phosphorylated BES1 relative to the total amount of BES1 for the three tested BRs at different concentrations. As shown in Figure 2b, BL exhibited the strongest effect on changing the phosphorylation status of BES1, followed by 28-HBL and 24-EBL. It is important to note that the appearance of the un/de-phosphorylated BES1 is the result of dephosphorylation of the pre-existing phosphorylated BES1 and the newly synthesized unphosphorylated BES1. We concluded that a plot of concentration-dependent accumulation of un/de-phosphorylated BES1 relative to the total amount of BES1 is a better way to evaluate the biological activities of synthetic BRs.

### 2.3. High Concentrations of 28-HBL Inhibit Root Growth 

Exogenous application of BRs has a dose-dependent effect on root growth, with growth-promoting effects at lower concentrations and growth-inhibiting effects at higher concentrations [26,50]. To compare the physiological activities of the three synthetic BRs, we adopted the Arabidopsis root growth assay [51]. As shown in Figure 3a,b, root length was slightly increased in response to a lower concentration (0.001 nM) of all three synthetic BRs. Both 28-HBL and BL strongly inhibited the root growth at concentrations from 0.01 nM to 100 nM. By contrast, 0.01 nM 24-EBL did not inhibit but rather stimulated the root growth. The 1 nM concentration seems to better differentiate the three BRs, as 28-HBL exhibited a much stronger inhibition of root elongation than 24-EBL, but not as strong as BL (Figure 3b). At concentrations greater than 10 nM, all three BRs showed the maximum effects on root growth inhibition. According to these findings, 28-HBL exhibited an intermediate activity in inhibiting the root growth between BL and 24-EBL. Similar to what we found with the BES1 dephosphorylation assay, a dose–response curve is needed to carefully evaluate the physiological activity of various synthetic BRs.

### 2.4. 28-HBL Is Stronger than 24-EBL in Rescuing the det2 Dwarfism in the Light

Synthetic BRs coupled with BR-deficient mutants not only confirmed the crucial roles of BRs in regulating various aspects of plant growth and development but also elucidated the BR biosynthetic pathway. The Arabidopsis BR-deficient mutants, *det2* and *cpd*, were originally discovered as light-signaling mutants based on their light-grown morphology in the dark [23,24]. The *det2-1* mutant, harboring a Glu-204-Lys substitution in DET2, shows a dwarf phenotype (Figure 4a), with smaller and darker green leaves, reduced male fertility, and delayed senescence and flowering compared to wild-type plants when grown in the light [23,52]. When grown in the dark, *det2* exhibited short and thick hypocotyls, expanded cotyledons, and developed primary leaf buds (Figure 5a) [52]. Its sequence homology with mammalian steroid 5α-reductases and metabolic analyses of endogenous BRs and exogenously applied radioisotope-labelled synthetic BRs concluded that DET2 is involved in reducing the double bond in the B-ring of the sterol core structure [23,53,54].

It was discovered that the exogenous application of BL could reverse the phenotypes of both light- and dark-grown *det2* [23,53]. We reasoned that the phenotype rescue of the *det2* mutant could be a better system to compare the biological activities of synthetic BRs because the wild-type Arabidopsis seedlings contain various active BRs that accumulate to different levels in different tissues or under different growth conditions [55,56,57]. We first grew the *det2-1* seedlings on the ½ MS medium containing different concentrations of BL, 24-EBL, or 28-HBL in the light for 20 days and subsequently examined their growth morphology. As shown in Figure 4a,b, at 1 nM concentration, both BL and 28-HBL caused a slight leaf expansion in light-grown *det2* mutants whereas 24-EBL had no observable impact on leaf growth. At 10 nM, BL rescued 95.8% of *det2* seedlings measured by petiole elongation and overall seedling size, 28-HBL resulted in significant partial rescue of the *det2* dwarfism, whereas 24-EBL exhibited a fairy weak partial *det2*-rescuing activity. When the BR concentration reached 20 nM or higher, BL completely rescued the *det2* dwarf phenotype while 28-HBL resulted in 76–90% fully rescued seedlings and 10–20% partially rescued *det2* seedlings. By contrast, no fully rescued *det2* seedlings were observed on the 24-EBL-containing medium and 100 nM 24-EBL caused ~80% of treated *det2* seedlings to exhibit a partially rescued morphology. The examination of the seedlings’ rosette morphology coupled with the quantitative analysis of the percentage of fully/partially rescued *det2* dwarfism clearly demonstrated that 28-HBL is approximately 2.5-fold less active than BL but is much stronger than 24-EBL in rescuing the light-grown growth defect of the *det2* mutant.

### 2.5. 28-HBL Is as Active as BL in Rescuing the det2 Phenotype in the Dark

BR regulates skotomorphogenesis (development in the dark), and BR-deficient and insensitive mutants often exhibit constitutive photomorphogenesis in the dark with short hypocotyl and opened cotyledons [58]. To compare the biological activities of the three tested synthetic BRs, we germinated the *det2* seeds and grew the resulting seedlings on the ½ MS medium containing varying concentrations of BRs in the dark and analyzed their hypocotyl length. As shown in Figure 5a, at 1 nM concentration, the hypocotyl length of *det2* seedlings grown on BL was ~50% that of the wild-type seedlings, representing the highest growth-promoting effect among the three tested synthetic BRs. At a concentration of 10 nM or higher, both BL and 28-HBL caused the *det2* hypocotyl to reach the same height as that of the wild-type seedlings grown on the normal ½ MS medium. It is interesting to note that high concentrations (50 and 100 nM) of BL and 28-HBL slightly suppressed the hypocotyl elongation (Figure 5a,b). By contrast, the *det2* hypocotyl elongation was only weakly enhanced by 24-EBL between 10 nM and 100 nM, with the *det2* hypocotyl length on the 100 nM 24-EBL-containing medium being shorter than that on 10 nM BL or 10 nM 28-HBL. Taken together, these results indicate that 28-HBL exhibited the same activity as BL in rescuing the short hypocotyl phenotype of the *det2* mutant while 24-EBL exhibited less than 10% of the biological activities of BL and 28-HBL.

### 2.6. 28-HBL Is Much Weaker than BL in the Rice Lamina Inclination Test

The rice lamina inclination test (RLIT) was originally developed to measure auxin activity [59] but became one of the most sensitive and widely used bioassays to measure BR activity [60]. As a matter of fact, RLIT (*Oryza sativa* L. cv Arborio J1) was used by Japanese scientists to compare the bioactivity of 28-HBL with BL and 24-EBL right after its successful chemical synthesis in 1983 [43], which concluded that 28-HBL was as active as BL and ~10-fold more active than 24-EBL. However, two other organic chemistry laboratories used other bioassays to measure the biological activities of their newly synthesized 28-HBL but concluded that 28-HBL was much weaker than BL [40,44].

We also performed a RLIT to compare the biological activities of the three synthetic BRs using the japonica rice cultiva Zhonghua 11 (*Oryza sativa* subsp. japonica). As shown in Figure 6a,b, under our experimental conditions, the second-leaf lamina joint angle of the rice Zhonghua 11 is about 75.7°. The addition of BRs into the incubation buffer strongly stimulated the rice lamina inclination. At a concentration of 1 μM, all three tested BRs dramatically increased the angle between the rice second-leaf lamina and the sheath to ~126°–131°. However, at 0.01 μM concentration, BL enlarged the lamina–sheath angle to 114.1°, whereas 28-HBL and 24-EBL only slightly enlarged the lamina joint angle to 96.3° and 91.8°, respectively. Thus, although 28-HBL could produce a strong stimulatory effect on the rice lamina inclination at high concentrations, it exhibited a similar weak activity as 24-EBL at lower concentrations. This finding is quite different than the first RLIT result reported for 28-HBL [43] but is consistent with recent findings showing the bioactivities of BRs could be influenced by different varieties/cultivars and bioassay protocols [42].

## 3. Discussion

Exogenous application of synthetic BRs has been used in agriculture to increase crop yields or to enhance the plant tolerance against a wide range of environmental stresses. The direct application of BRs in the field is regarded as an economically and eco-friendly approach and has thus gained great interest over the years. This, together with the fact that it is relatively easy to chemically synthesize, promotes the industrial-scale production of 28-HBL. However, the quality of different batches of chemically synthesized 28-HBL, even from the same manufacturer, varies due to variability in the ratio of active 22*R*,23*R*-28HBL and its relatively inactive stereoisomer 22*S*,23*S*-28HBL in the final products that reach farmers [61]. This is largely due to choices of the chiral ligands/catalysts used to catalyze the asymmetric hydroxylation of the 22,23-double bond of the steroid side chain and the costly purification scheme needed to enrich the 22*R*,23*R*-28-HBL [36,42,61]. Thus, it is essential to establish a standardized bioassay system to test the bioactivity and quality of freshly synthesized 28-HBL (and other BRs or their analogs). This will ensure the full potential of 28-HBL/BRs utilized to enhance agricultural production or stress tolerance.

RLIT is commonly used to detect BRs’ bioactivity because BRs have a high capacity to induce lamina inclination between the plant leaf blade midrib and the vertical stem. However, the use of different rice cultivars is known to result in different bioactivity of 28-HBL. Two Japanese groups reported different bioactivity of 28-HBL relative to BL right after their successful synthesis of 28-HBL, with one group claiming similar bioactivity of 28-HBL and BL and the other group revealing a rather weaker bioactivity of 28-HBL relative to BL [40,44]. In the current study, RLIT using the japonica cultivar ZH11 revealed that 28-HBL was slightly more active than 24-EBL but weaker than BL to induce the lamina inclination at lower concentrations (Figure 6). The results reported here provide additional support for an earlier claim that although RLIT was widely used, different laboratories produced variable results due to different cultivars/strains and assay protocols used, making it invalid for comparison of BR bioactivity from published works [42]. *Arabidopsis thaliana* is a powerful system used to test BRs’ bioactivity due to well-established protocols, reagents, and biomarkers of BR responsiveness. The wild-type *Arabidopsis thalian* strain and its BR-deficient mutant *det2* are widely available in the scientific community around the globe. Importantly, as a small plant, the whole Arabidopsis seedlings were convenient for performing BR treatment and for observing changes in growth and/or morphological changes of leaves, hypocotyls, petioles, and roots. Thus, bioassays using whole Arabidopsis seedlings could become a standard bioassay system to evaluate bioactivities of every batch of newly synthesized 28-HBL or other BR analogs of different manufacturers at multiple levels.

Here, the bioactivity of 28-HBL was systematically explored and compared to the widely used BRs BL and 24-EBL by performing a series of bioassays using whole Arabidopsis seedlings, especially a BR-deficient mutant, removing the unclear results produced using different plant species or explant segments. Its ability to trigger the BR signal transduction and regulate plant growth has been analyzed in detail. In the well-established BR signaling pathway, the two key transcription factors BES1 and BZR1 are primarily phosphorylated by the activated protein kinase BIN2 (BRASSINOSTEROID INSENSITIVE2) [62] when the BR is absent. In the presence of BR, BR signal transduction is elicited. BZR1 and BES1 are dephosphorylated and activated to control the expression of thousands of BR-responsive genes [62]. Because there are several putative copies of BIN2 phosphorylation sites in BES1/BZR1 [63], the dephosphorylated and phosphorylated BES1 can be easily separated and detected by SDS-PAGE followed by an immunoblot analysis using an anti-BES1 antibody. Based on these, the current study reported the high bioactivity of 28HBL to induce BR signaling. Higher concentrations of 28-HBL, BL, or 24-EBL strongly induced an accumulation in dephosphorylated BES1 and a reduction in phosphorylated BES1 (Figure 2), causing an alteration of the transcript abundance of the BR-responsive genes (Figure 1). These results demonstrate that high doses of the three tested BRs exhibited similarly high biological activity. To further explore the variation in bioactivity, lower doses of BRs were applied in the bioassays. At lower concentrations, 28-HBL exhibited stronger activity to elicit the BR response than 24-EBL, as evidenced by higher proportional amounts of dephosphorylated BES1 (Figure 2) and higher fold changes of BR-responsive gene expression by 28-HBL (Figure 1). In addition to the molecular and biochemical analyses, physiological studies have directly showed the high capacity of 28-HBL to regulate plant growth. It has been reported that low concentrations of BRs can stimulate the elongation of vegetative tissues, including hypocotyls and epicotyls [64]. Nanomolar levels of 28-HBL caused pronounced elongation of the hypocotyl of dark-grown *det2* (Figure 5) and the petiole of light-grown *det2* (Figure 4). 28-HBL showed a slightly lower ability than BL but an extremely greater ability compared to 24-EBL to normalize the phenotype of light-grown *det2* and a comparable ability with BL to rescue the phenotype of dark-grown *det2*. These reversals of the *det2* seedlings in the light and dark further revealed that 28-HBL had a high BR bioactivity. Additionally, 28-HBL regulated root growth in a dose-dependent manner, as BL does, with growth promotion at lower concentrations and growth inhibition at higher concentrations (Figure 3), implying its high capacity to regulate root growth. These outstanding performances of 28-HBL to trigger the BR response and regulate plant growth probably explain its positive influence on crop plants. Using whole Arabidopsis seedlings, the molecular, biochemical, and physiological bioassays consistently demonstrated the strong bioactivity of 28-HBL in the current study. This makes it valid to utilize this multi-level bioassay system to estimate the convincing bioactivities of newly prepared 28-HBL/BRs and their analogs prior to their widespread application.

The similar or different bioactivities of various BRs depend on their similarity and variation in structures. Several bioassays have revealed the key structural features, such as the functional groups on the basic BR backbone and the stereochemistry present in the A- and B-rings and the side chain, which are crucial for high biological activities and variants (Appendix A) [42]. BL, 24-EBL, and 28-HBL share a common cholestane skeleton and the same 2α, 3α-vicinal hydroxyls in the A-ring, in which 2α hydroxyl is essential for greater bioactivity, as well as 7-oxalactone in the B-ring, which provides higher biological activity than 6-oxalactone types and non-oxidized BRs [42]. Additionally, BL, 28-HBL, and 24-EBL contain a vicinal 22*R*, 23*R* diol structure in the side chain, which appears to be essential for high bioactivity [42]. These structure–activity relationships are strongly supported by the experimental findings in the current study, as evidenced by the high biological activity of BL, 28-HBL, and 24-EBL. The structures of the three BRs differ in the alkyl-substitutions in the side chain at C-24 or the configuration at C-24 (Appendix A), which contribute to their differences in bioactivity. The *S* configuration was more crucial for high activity, as evidenced by the presence of an *S*-methyl group at C-24 in BL, the most active BR that is synthesized from campesterol, as opposed to an *R*-methyl group at C-24 in 24-EBL, which has reduced bioactivity. The most active C29 BR, 28-HBL that is derived from sitosterol, bears an *S*-ethyl at C-24. The current study clearly showed that 28-HBL had a higher bioactivity than 24-EBL and a reduced bioactivity compared to BL when lower doses of BRs were applied. Thus, the predicted structure–activity relationships are in agreement with the current comparative details of the bioactivities of three BRs. This provides strong evidence that the multi-level bioassay system using whole Arabidopsis seedlings is feasible to accurately evaluate BRs’ bioactivity.

The variable structure of BRs caused variations in the binding affinity with their receptor BRI1, leading to the variable capacity to elicit the BR signal transduction. Based on the published crystal structure of the BL-bound extracellular domain of BRI1, it appears that the 24*S*-methyl group interacts with the Tyr-597 and Leu-615 residues (Appendix A) [65]. The replacement of the 24*S*-methyl group with a 24*S*-ethyl group slightly affects 28-HBL–BRI1 binding. However, the replacement of the 24*S*-methyl group with a 24*R*-methyl group likely pushes the terminal isopropyl group (C25-C26/C27) into the free space of Tyr-597/Leu-615, which could significantly reduce the binding affinity of 24-EBL with BRI1. Therefore, BL exhibits the greatest binding affinity to BRI1 of the three BRs, followed by 28-HBL. This elucidates the fact that 28-HBL has a higher capacity than 24-EBL to induce the BR response and control plant growth and should facilitate the understanding and the design of potent BR analogs. However, Tyr-597 and Leu-615 are conserved among BRI1 homologs of representative monocotyledon and dicotyledon species (Appendix A). Thus, it is unlikely that these two residues are responsible for the differential bioactivities of 28-HBL observed in the different bioassays. Nevertheless, the surrounding residues of BRI1 in different plant species could alter the spatial positioning of the two BR-interacting residues, thus altering the binding affinity of BRI1 with 28-HBL. Alternatively, the different bioactivities of BL, 24-EBL, and 28-HBL might be at least partially due to the presence or absence and different levels of BR-inactivating enzymes [57,66,67,68] in different plants or in the same plant grown under different environmental conditions.

## 4. Materials and Methods

### 4.1. Plant Materials and Growth Conditions

Seeds of wild-type (Col-0) *Arabidopsis thaliana* and *det2* were surface sterilized using the ethanol-washing protocol [26] with some modifications. Seeds were surface sterilized by incubation with 70% (*v*/*v*) ethanol for 10 min, followed by two 5 s washes with purity ethanol. Seeds were then dried on sterilized filter papers under sterile conditions and sown onto a half-strength Murashige and Skoog medium containing 1% (*w*/*v*) sucrose (Murashige and Skoog salt was purchased from Coolaber, Beijing, China) alone or with different concentrations of BRs. After a dormancy break at 4 °C for 48 h, seeds were grown under a long-day photoperiod (16 h light/8 h dark) or in constant darkness at 22 °C.

### 4.2. Phytohormone Preparation

BL and 24-EBL were purchased from Wako (Tokyo, Japan) and Sigma (Saint Louis, MO, USA), respectively. 28-HBL was provided by Windeal (Nanchang, China). A stock solution of 2 mM BL, 24-EBL, or 28-HBL was prepared by dissolving each compound in pure ethanol. The working solutions of BRs required in different bioassays were diluted in double-distilled water.

### 4.3. RT-qPCR of BR-Responsive Genes

10-day-old wild-type (Col-0) Arabidopsis seedlings grown on a half-strength MS medium were transferred to a liquid half-strength MS medium containing 0.1 or 1 μM BL, 24-EBL, or 28-HBL and incubated for 2 h. The seedlings were harvested, ground into a fine powder in liquid nitrogen, and their total RNA was extracted using an RNeasy Plant Mini Kit (Vazyme, Nanjing, China). One microgram of each total RNA sample was reverse transcribed into first-strand cDNA using a HiScript III Strand cDNA Synthesis Kit (Vazyme, Nanjing, China). The cDNA was subjected to a quantitative PCR (qPCR) analysis with gene-specific oligonucleotides listed in Appendix A. The qPCR assays were performed with a CFX96 Real-Time System (Bio-Rad, Hercules, CA, USA) and ChamQ Universal SYBR qPCR Master Mix (Vazyme, Nanjing, China) following the manufacturer’s instructions.

### 4.4. Analysis of BR-Induced BES1 Dephosphorylation

Ten-day-old wild-type (Col-0) Arabidopsis seedlings were incubated in the liquid half-strength MS medium alone or with 0.001, 0.01, 0.1, or 0.5 μM BL, 24-EBL, or 28-HBL to trigger BR-induced BES1 dephosphorylation. In total, 30 mg of seedling materials were quickly frozen in liquid nitrogen and ground into a fine powder. Total proteins were isolated from the samples by adding 2 × SDS loading buffer (100 mM Tris-HCl, pH 6.8, 20% [*v*/*v*] glycerol, 4% [*w*/*v*] SDS, 0.02% [*w*/*v*] Bromophenol Blue, and 100 mM DTT), heating at 95 °C for 10 min, and immediately centrifuging the samples at 10,000× *g* at room temperature for 10 min. The supernatants were collected, immediately separated on a 12% SDS-PAGE gel, and analyzed by an immunoblot with antibodies generated against BES1. The His–BES1 fusion protein, which encompasses residues 88 to 335 of the full-length BES1, was expressed in *E. coli* and followed by affinity purification using a Co^2+^ Resin (GE Healthcare, Chicago, IL, USA). Then, the purified His–BES1 proteins were used to make the custom antibody and the purified GST-BES1 was used to affinity purify anti-BES1 antibodies from the anti-BES1 serum. The signals in the immunoblot were detected with horseradish peroxidase-linked secondary antibodies by chemiluminescence (Invitrogen, Carlsbad, CA, USA).

### 4.5. Root Elongation Inhibition Assay

Wild-type (Col-0) Arabidopsis seeds were surface sterilized with 1% (*v*/*v*) NaClO for 15 min, washed three times with sterile deionized water, and sown on the half-strength MS medium containing 0.8% (*w*/*v*) agar and 1% (*w*/*v*) sucrose alone or with 0.001, 0.01, 0.1, 1, 10, or 100 nM BL, 24-EBL, or 28-HBL. After vernalization for 2 days at 4 °C, seedlings were grown vertically for 9 days in the light. After taking photographs, root length was measured, and a statistical analysis was carried out using ImageJ software (version 1.52a, Bethesda, MD, USA).

### 4.6. Phenotypic Rescue of det2 in the Light or Dark

*det2* seeds were surface sterilized with 1% (*v*/*v*) NaClO for 15 min, washed three times with sterile deionized water, and sown on the half-strength MS medium containing 0.8% (*w*/*v*) agar and 1% (*w*/*v*) sucrose alone or with 1, 10, 20, 50, 80, or 100 nM BL, 24-EBL, or 28-HBL. The plates were vernalized for 2 days at 4 °C in the dark and transferred to a growth chamber. Following 20 days of growth in the light or 5 days of growth in the dark, the plants or seedlings were photographed. The hypocotyl length of dark-grown seedlings was measured, and a statistical analysis was carried out using ImageJ software (version 1.52a, Bethesda, MD, USA).

### 4.7. Rice Lamina Inclination Test

The rice lamina inclination test was performed as previously described with some modifications [69]. The rice seeds (*Oryza sativa* L. ssp. Japonica cv. Zhonghua 11) were soaked in water at 28 °C for 2 days, followed by the transfer into a growth chamber at 37 °C for an additional 1 day until the seeds germinated with a radicle length of approximately 1 mm. The germinated seeds were placed into the wells of a 96-well PCR plate and grown in the hydroponics setup. After approximately 8 days of growth, approximately 2 cm long segments containing the second-leaf lamina joint, leaf blade, and leaf sheath were cut off and incubated in sterile water for 10 min. Then, the segments were transferred into Petri dishes containing 0, 0.01, or 1 μM BL, 24-EBL, or 28-HBL, respectively, and incubated at 28 °C in the dark for 2 days. The photographs of these segments were taken, and the lamina joint angles were measured with ImageJ software (version 1.52a, Bethesda, MD, USA).

### 4.8. Sequence Analysis

The protein sequences of BRI1 homologs in Arabidopsis and other species were downloaded from TAIR (https://www.arabidopsis.org/) (Newark, CA, USA, accessed on 1 May 2022) and NCBI (https://www.ncbi.nlm.nih.gov/) (Bethesda, MD, USA, accessed on 1 May 2022). Sequence alignment of the island domains of 20 BRI1s from 20 species was performed using CLUSTALW (https://www.genome.jp/tools-bin/clustalw) (Uji, Japan, accessed on 1 May 2022).

### 4.9. Statistical Analysis

The results are expressed as means ± SD. For RT-qPCR, the rescue of the hypocotyl phenotype of dark-grown *det2*, and the rice lamina inclination test, statistical analyses were performed using a one-way analysis of variance (ANOVA). Otherwise, a two-way analysis of variance (ANOVA) was employed. Subsequent post-hoc analyses were performed using the Tukey’s method. Values of *p* < 0.05 were considered significant for all analyses. All analyses were performed using GraphPad Prism (version 8.0.2, La Jolla, CA, USA).

## Figures and Tables

**Figure 1 ijms-24-09377-f001:**
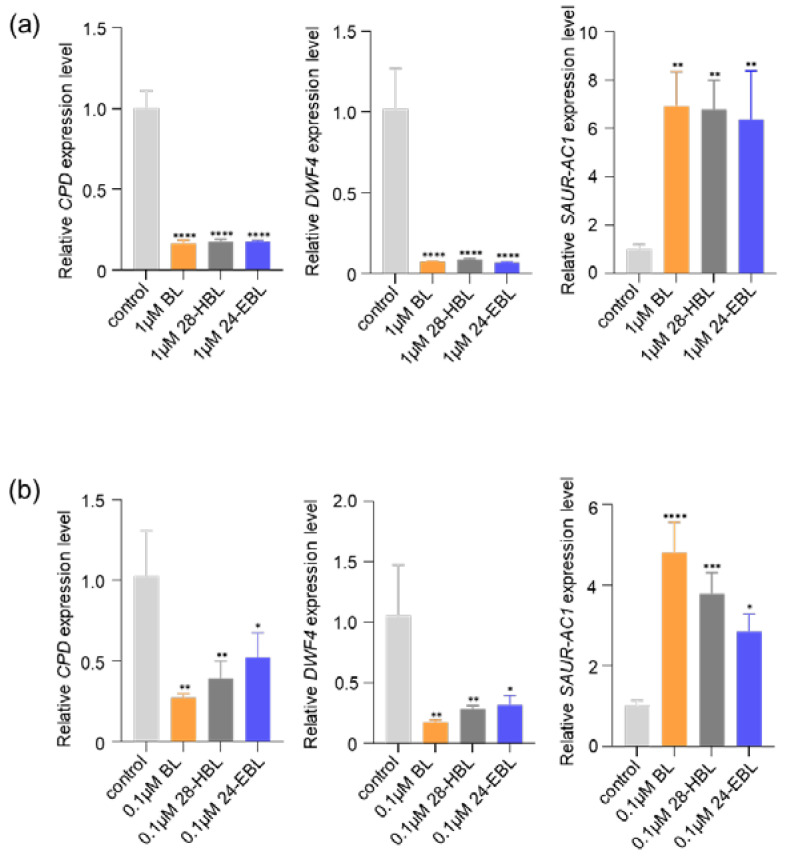
Differential impacts of BL, 28-HBL, and 24-EBL on the transcript abundance of three well-known BR-responsive genes. (**a**) RT-qPCR analysis of the transcript levels of *CPD*, *DWF4*, and *SAUR-AC1* in wild-type Arabidopsis seedlings treated with 1 μM BL, 28-HBL, or 24-EBL. (**b**) RT-qPCR analysis of the transcript abundance of *CPD*, *DWF4*, and *SAUR-AC1* in wild-type Arabidopsis seedlings treated with 0.1 μM BL, 28-HBL, or 24-EBL. In (**a**,**b**), three independent biological replicates were carried out. The seedlings treated with solutions containing the same concentration of ethanol as the BR solutions were used as controls. Asterisks indicate significant differences from the corresponding control (*, *p* < 0.05; **, *p* < 0.01; ***, *p* < 0.001; ****, *p* < 0.0001). Error bars denote ± SD.

**Figure 2 ijms-24-09377-f002:**
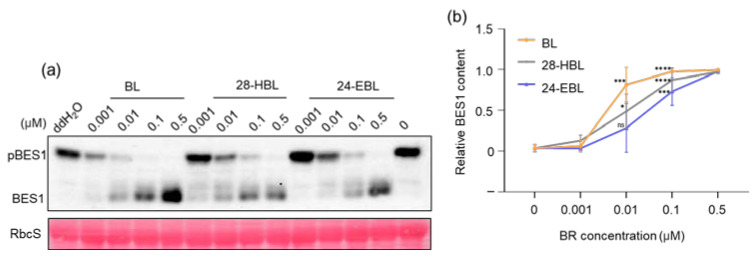
Differential effects of BL, 28-HBL, and 24-EBL on the phosphorylation status of BES1. (**a**) Immunoblot analysis of BES1 in wild-type Arabidopsis seedlings treated with different concentrations of BL, 24-EBL, or 28-HBL. Total protein extracts of 10-day-old Arabidopsis seedlings treated with different concentrations of BRs were separated by SDS-PAGE and analyzed by immunoblot with an anti-BES1 antibody. The ponceau red staining of the small subunit of the ribulose-1,5-bisphosphate carboxylase/oxegenase (RbcS) was used as the loading control. (**b**) The concentration-dependent changes of the ratio of un/de-phosphorylated BES1 to the total amount of BES1 (un/de-phosphorylated and phosphorylated BES1). Two biological repeats were performed and the intensity of BES1 bands was quantified by ImageJ software (version 1.52a, Bethesda, MD, USA). Asterisks indicate significant differences from the corresponding control (*, *p* < 0.05; ***, *p* < 0.001; ****, *p* < 0.0001; ns, no significance). Error bars denote ± SD.

**Figure 3 ijms-24-09377-f003:**
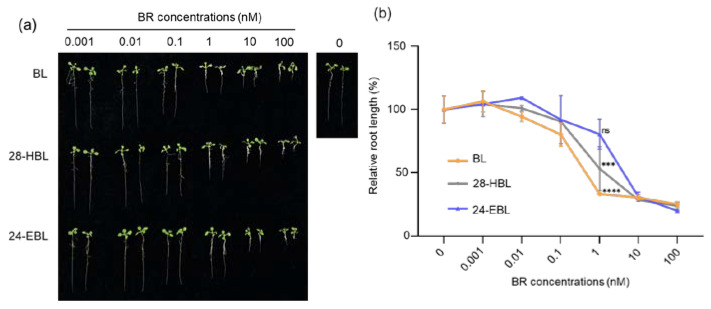
BL, 28-HBL, and 24-EBL exhibit different activities on the root growth of wild-type Arabidopsis seedlings. (**a**) Photographs of 9-day-old wild-type Arabidopsis seedlings grown vertically on half-strength MS medium containing 0.001, 0.01, 0.1, 1, 10, or 100 nM BL (top row), 28-HBL (middle row), or 24-EBL (the bottom row). (**b**) Quantitative analysis of the impacts of the three BRs on root growth of the wild-type Arabidopsis seedlings. Nine-day-old light-grown seedlings were carefully removed from Petri plates and photographed, and their root lengths were measured using ImageJ. Each data point indicates the average root lengths of ~60 seedlings from two independent biological replicates (n = 30 for each replicate), expressed as the relative value compared to the root length of wild-type seedlings grown on ½ MS medium containing the same volume of ethanol used to make various BR solutions. Asterisks indicate significant differences from the corresponding control (***, *p* < 0.001; ****, *p* < 0.0001; ns, no significance). Error bars denote ± SD.

**Figure 4 ijms-24-09377-f004:**
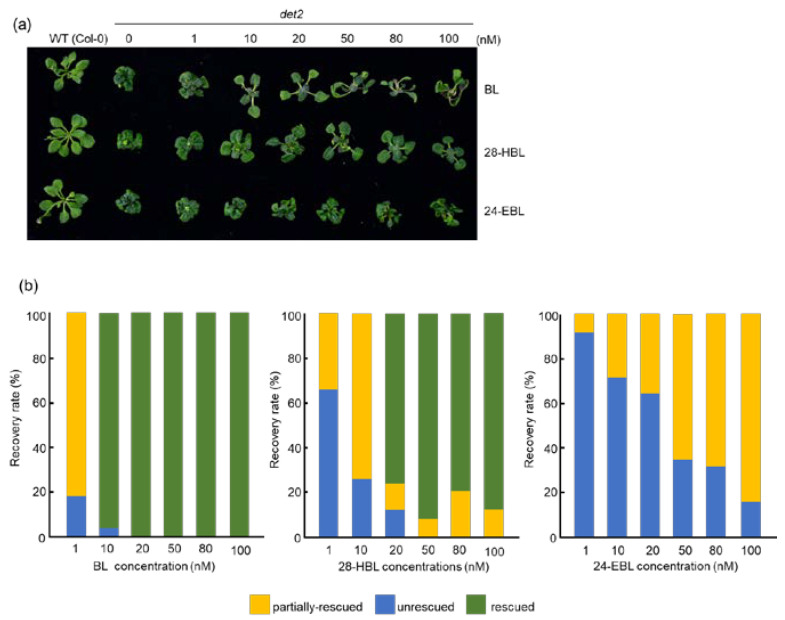
Phenotypic rescue of light-grown *det2* seedlings by BL, 28-HBL, or 24-EBL. (**a**) Photographs of 20-day-old light-grown *det2* seedlings on ½ MS medium without or with 1, 10, 20, 50, 80, or 100 nM BL (the upper row), 28-HBL (the middle row), or 24-EBL (the bottom row). (**b**) Quantitative analysis of rescued phenotypes of *det2* using different concentrations of BRs indicated in (**a**). Two independent biological replicates were carried out (n = 23 for each replicate).

**Figure 5 ijms-24-09377-f005:**
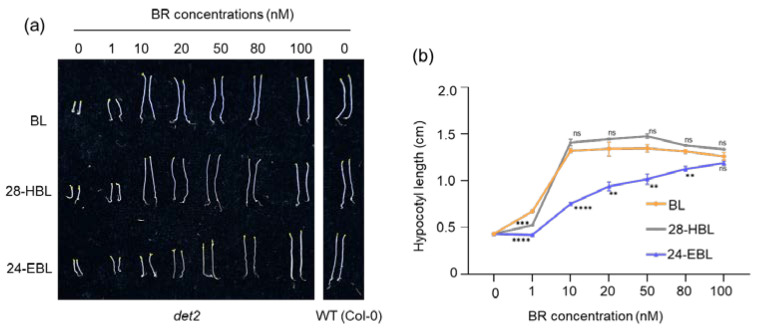
Rescue of the *det2* short hypocotyl phenotype in the dark by BL, 28-HBL, and 24-EBL. (**a**) Photographs of 5-day-old dark-grown *det2* seedlings on half-strength MS medium without or with 1, 10, 20, 50, 80, or 100 nM BL, 28-HBL, or 24-EBL. (**b**) Quantitative analysis of the sensitivity of dark-grown *det2* to varying concentrations of different BRs indicated in (**a**). Five-day-old dark-grown seedlings were carefully removed from Petri plates and photographed, and their hypocotyl lengths were measured using ImageJ. Two independent biological replicates were carried out (n = 45 for each replicate). Asterisks indicate significant differences from the treatment by the same concentration of BL (**, *p* < 0.01; ***, *p* < 0.001; ****, *p* < 0.0001; ns, no significance). Error bars denote ± SD.

**Figure 6 ijms-24-09377-f006:**
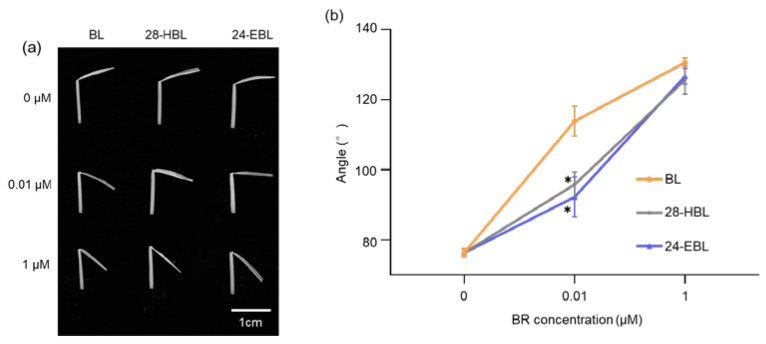
The rice lamina inclination test to compare bioactivities of BL, 28-HBL, and 24-EBL. (**a**) Representative leaf segments containing the lamina joint, blade, and sheath from the WT rice plants (japonica cultiva Zhonghua 11) after 2-day dark incubation in the solution with different concentrations of BL, 28-HBL, or 24-EBL. (**b**) Quantitative analysis of the lamina inclination in response to different concentrations of BRs. The lamina–sheath angles were measured by ImageJ. Two independent biological replicates were carried out (n > 20 for each replicate). Asterisks indicate significant differences from the treatment by the same concentration of BL (*, *p* < 0.05). Error bars denote ± SD.

## Data Availability

Not applicable.

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
