# Peer review of "Assessment of Biological Activity of 28-Homobrassinolide via a Multi-Level Comparative Analysis"

_ijms, 2023, doi:10.3390/ijms24119377_

Round 1
Reviewer 1 Report
Dear authors, I did not find any serious errors in your article, but unfortunately, I have to state that the mentioned article does not bring any new knowledge to the scientific community. The biological effects of the tested BRs have been known for a long time and are well documented, as well as the chemical synthesis of these derivatives has already been described. BL derivatives (24-epiBL and 28-homoBL) are widely used in agriculture, to increase production or increase resistance to stresses. Also, all the mentioned methods/bioassays are commonly used. Therefore, I cannot recommend it for publication.
Minor points:
Abbreviations should not be used in the title of article.
The control should contain the same solvent concentration as for the tested substances (i.e. not only ddH2O, but ddH2O + ethanol).
Are the Anti-BES1 antibodies used commercially available or did you prepare them yourself? If so, you should describe the characteristics of these antibodies.
Author Response
Dear reviewer,
Thank you very much for the the time and effort that you dedicated to providing feedback on our manuscript. We are grateful for the insightful comments that helped improve our paper. We have made modifications to the revised manuscript based on the professional suggestions made by the reviewers. Those changes and additions are highlighted in yellow within the revised manuscript. Please see below, in blue, for a point-by-point response to your kind comments and concerns.
Reviewers' Comments to the Authors:
Dear authors, I did not find any serious errors in your article, but unfortunately, I have to state that the mentioned article does not bring any new knowledge to the scientific community. The biological effects of the tested BRs have been known for a long time and are well documented, as well as the chemical synthesis of these derivatives has already been described. BL derivatives (24-epiBL and 28-homoBL) are widely used in agriculture, to increase production or increase resistance to stresses. Also, all the mentioned methods/bioassays are commonly used. Therefore, I cannot recommend it for publication.
Response: While we appreciate the reviewer’s feedback, we respectfully disagree. We think this study makes a valuable contribution to the study and modern agriculture. As we know, chemically synthesized BRs are widely used in agriculture due to their ability to promote yield and enhance tolerance to stress. Though a famous natural BR, the certainty of the biological activity of 28-HBL has not been consistently reported. Additionally, the choices of chiral ligands/catalysts and purification schemes lead to the variable ratios of 22S,23S-28HBL and 22R,23R-28HBL in the final products, which affect the quality and bioactivity of industrially produced 28-HBLs. Though several assays have been used to assess the bioactivities of BRs and their analogs, frequently different labs obtain various and inconsistent results, probably due to the different protocols, and plant explants and species used. Thus, the assessment of biological activities of newly prepared 28-HBL (including the synthetic BR analogs) was required to confirm the full potential of this fascinating growth regulator used in modern agriculture. Based on these, we try to remind academic and industrial laboratories to use standard bioassay systems to evaluate BRs’ bioactivity accurately, ensuring the full potential of BRs used in modern agriculture.
Comment for Minor points:
- Comment: Abbreviations should not be used in the titleof article.
Response: Thank you for pointing this out. All the abbreviations have been replaced by the full name in the title of our revised manuscript.
- Comment: The control should contain the same solvent concentration as for the tested substances (i.e. not only ddH2O, but ddH2O + ethanol).
Response: Thank you for your professional suggestion. Yes, you are right. In fact, the solution used for the control samples contained the same solvent concentration as for the substances in our experiments. We have corrected in our revised manuscript. Please see page 4, lines 168-170.
- Comment: Are the Anti-BES1 antibodies used commercially available or did you prepare them yourself?If so, you should describe the characteristics of these antibodies.
Response: Thank you for your advice. We prepared the anti-BES1 antibodies ourselves, and we have described the antibodies in detail on page 12, lines 528-532 in our revised manuscript.
Reviewer 2 Report
The manuscript is well-written, I recommend it to accept.
Author Response
Dear reviewer,
Thank you very much for the time and effort that you dedicated to providing feedback on our manuscript. We have made modifications to the revised manuscript based on the professional suggestions made by the other reviewers. Those changes and additions are highlighted in yellow within the revised manuscript.
Reviewer 3 Report
General comment:
This paper is study about a variety of bioassays at the molecular, biochemical, and physiological levels to systematically and accurately assess 28-HBL’s bioactivity. Overall, this paper has well covered the authors’ findings on the bioactivity of 28-HBL and 24-EBL. However, authors need to perform critical analysis and interpret all these studies and come up with a conclusion for each section. It’s good that the author had this finding written but readers would preferably want to know what had the authors concluded from all these studies, instead of what the author of the literature studies had concluded. The title of this paper needs improvement as well, the current title sounds very unprofessional. To conclude, this paper needs to revise it carefully before it can be considered in high impact journal. Hope below comments will be able to help to further improve the paper.
Specific comment:
Abstract:
- The current abstract sounds more like an introduction to brassinolide rather than this paper, an abstract should summarize the whole paper, please revise the abstract
- Please do not use the terms “we, our” in a scientific paper, change to passive form instead
- The abstract is not properly wrapped up, please include a small conclusion at the end of the abstract
Introduction:
- Revised Introduction section based on the structure below:
1st paragraph: Problem statement
2nd paragraph: Current ongoing solution
3rd paragraph: Proposed solution in this work.
4th paragraph: Summarized the current research novelty and objective of this work.
- Introduction should be covered the gap of the research. However, it is not well covered in this section.
- Also, please mention the important of this study to society as well as industry.
- Include more usage of BRs in the introduction and why it is needed in higher concentration
- Authors can consider including some applications of synthetic BRs in the industry
- The problem statement is not strong enough, please enhance the problem statement
- Suggest the authors to read some additional materials:
“Interplay Impact of Exogenous Application of Abscisic Acid (ABA) and Brassinosteroids (BRs) in Rice Growth, Physiology, and Resistance under Sodium Chloride Stress” and “Brassinosteroids promote starch synthesis and the implication in low-light stress tolerance in Solanum lycopersicum”
- Briefly describe some of the uses of 24-EBL/28-HBL rather than just mentioning them
- Please include a summary of novelty of this paper in the introduction
- Authors are advised to include more information on the main species studied, Arabidopsis. The following paper may aid in the writing of this paper: “Transcription Factor ChbZIP1 from Alkaliphilic Microalgae Chlorella sp. BLD Enhancing Alkaline Tolerance in Transgenic Arabidopsis thaliana”
- The aim and objectives of this paper are not clearly stated, please revise
- Please do not include any results in the introduction section
Material and Method:
- Please ensure that the arrangement of the sections follow the author guidelines
- Please briefly describe the protocol rather than only mentioning what protocol were used
- Please mention why is the experiment carried out under long-day photoperiod conditions
Body:
Results and Discussion
- Kindly improve on the discussion. What is the significance of the results of the work?
- Please give more authors’ opinion before including a literature reference as support, what does the authors think of the results and why does the results act so? After mentioning these then only place the citations to support the authors’ claims.
- Please link the results together, the authors did not describe what does the results mean and how are the results interrelated
- Please improve on figures, simple line graphs may not be considered as a complete figure for some journals
- Please revise the discussion section, the current discussion did not link the results with the discussion, report a result then discussion on that result, what are the significance of that results to the growth of the plant? The following papers may aid in the writing of the paper: “Mesorhizobium improves chickpea growth under chromium stress and alleviates chromium contamination of soil” and “The Effects of Biofertilizers on Growth, Soil Fertility, and Nutrients Uptake of Oil Palm (Elaeis Guineensis) under Greenhouse Conditions”
- Include more author insights on each results and discussion provided, this paper lacks author insights on each result obtained by the author
Conclusions
- Conclusion section is missing, please include a conclusion section to properly wrap up the whole manuscript
References
- Kindly revise reference format according to the author guideline.
- It is suggested to cite references within 5 years of research to maintain the reliability of results obtained.
- There are references found to be outdated.
Extensive editing of English language required
Author Response
Dear reviewer,
Thank you very much for the time and effort that you dedicated to providing feedback on our manuscript. We are grateful for the insightful comments that helped improve our paper. We have made modifications to the revised manuscript based on the professional suggestions made by the other reviewers. Those changes and additions are highlighted in yellow within the revised manuscript. Please see below for a point-by-point response to your kind comments and concerns.
General comment:
This paper is study about a variety of bioassays at the molecular, biochemical, and physiological levels to systematically and accurately assess 28-HBL’s bioactivity. Overall, this paper has well covered the authors’ findings on the bioactivity of 28-HBL and 24-EBL. However, authors need to perform critical analysis and interpret all these studies and come up with a conclusion for each section. It’s good that the author had this finding written but readers would preferably want to know what had the authors concluded from all these studies, instead of what the author of the literature studies had concluded. The title of this paper needs improvement as well, the current title sounds very unprofessional. To conclude, this paper needs to revise it carefully before it can be considered in high impact journal. Hope below comments will be able to help to further improve the paper.
Response: We would like to thank you for your careful reading, helpful comments, and constructive suggestions, which have significantly improved the presentation of our manuscript. Based on your suggestion and request, we have changed our title to “Assessment of biological activity of 28-homobrassinolide via a multi-level comparative analysis”. In the revision, we have reorganized the introduction, improved the discussion, and strengthened our conclusion and objectives. Meanwhile, the language pf the manuscript has been edited carefully, and we hope that our paper can be improved again.
Specific comment:
Comments for Abstract:
1、The current abstract sounds more like an introduction to brassinolide rather than this paper, an abstract should summarize the whole paper, please revise the abstract.
Response: Thank you for pointing out this problem in the manuscript. We have restructured the abstract in the revised manuscript to concisely summarize our work.
2、please do not use the terms “we, our” in a scientific paper, change to passive form instead.
Response: Thank you for pointing this out and we have used other presentation to avoid the terms “we, our”.
3、The abstract is not properly wrapped up, please include a small conclusion at the end of the abstract.
Response: Thank you for your helpful advice, and we have included the conclusion in the abstract on page 1, lines 27-30 in the revised manuscript.
Comments for Introduction:
- Revised Introduction section based on the structure below:
1st paragraph: Problem statement
2nd paragraph: Current ongoing solution
3rd paragraph: Proposed solution in this work.
4th paragraph: Summarized the current research novelty and objective of this work.
Response: Thank you for your constructive suggestions. We have reorganized the introduction section in the revised manuscript. And the additions are highlighted in yellow.
2、Introduction should be covered the gap of the research. However, it is not well covered in this section.
Response: Thank you for your suggestion, and we have strengthened to cover the gap of the research on page 3, lines 120-124 in the revised draft.
3、Also, please mention the important of this study to society as well as industry.
Response: Thank you for this advice, and we have mentioned the importance of this study to the modern agriculture and also included its importance to the industrial production of BRs. Please see page 3, lines 129-131 in the revised draft.
4、 Include more usage of BRs in the introduction and why it is needed in higher concentration.
Response: Thank you for your advice. We have mentioned that the effect of BRs on plant growth or stress tolerance depends on the appropriate dose and the correct stage of plant development on page 2 lines 68-70. Various concentrations of BRs are required to improve the crop trait and quantities, increase the crop yield and stress tolerance according to the various species and developmental stages.
5、Authors can consider including some applications of synthetic BRs in the industry.
Response: Thank you for your advice. It would have been interesting to explore this aspect. We have tried to mention the direct application of synthetic BRs in the industry. But we have noticed that most published papers have focused on its function in modern agriculture, and its direct function in industry has not been widely revealed.
6、The problem statement is not strong enough, please enhance the problem statement.
Response: Thank you for your advice, and we have enhanced the problem statement in our revised manuscript.
7、 Suggest the authors to read some additional materials:
“Interplay Impact of Exogenous Application of Abscisic Acid (ABA) and Brassinosteroids (BRs) in Rice Growth, Physiology, and Resistance under Sodium Chloride Stress” and “Brassinosteroids promote starch synthesis and the implication in low-light stress tolerance in Solanum lycopersicum”
Response: Thank you for your kind suggestions, and we have read these two papers and included one of the papers mentioned in the revised manuscript.
8、 Briefly describe some of the uses of 24-EBL/28-HBL rather than just mentioning them.
Response: Thank you for your suggestion. We have briefly described some of the uses of 24-EBL/28-HBL in the revised manuscript.
9、Please include a summary of novelty of this paper in the introduction.
Response: Thank you for your suggestion. We have strengthened the novelty of this paper in the introduction on page 3, lines 125-131 of the revised manuscript.
10、Authors are advised to include more information on the main species studied, Arabidopsis. The following paper may aid in the writing of this paper: “Transcription Factor ChbZIP1 from Alkaliphilic Microalgae Chlorella sp. BLD Enhancing Alkaline Tolerance in Transgenic Arabidopsis thaliana”
Response: Thank you for your advice. We have added the suggested content in the discussion section on Page 9, lines 397-404.
11、The aim and objectives of this paper are not clearly stated, please revise.
Response: Thank you for your advice, and we have strengthened the aim and objectives of the current study.
12、 Please do not include any results in the introduction section
Response: Thank you for your suggestion. We have deleted the “results” in the introduction section.
Comments for “Material and Method”:
- Please ensure that the arrangement of the sections follow the author guidelines.
Response: Thank you for your kind suggestions, and we have ensured that this section has followed the author guidelines.
- Please briefly describe the protocol rather than only mentioning what protocol were used.
Response: Thank you for your advice. Some brief descriptions of the protocols have been included on page 11, lines 495-500 and page 12, lines 528-532 in the revised manuscript.
- Please mention why is the experiment carried out under long-day photoperiod conditions.
Response: Thank you for your suggestion. Most of the Arabidopsis plants are grown well under long-day photoperiod or short-day photoperiod conditions in the published paper (except where special growth conditions were required). And under long-day photoperiod conditions, some growth- and development-retarded Arabidopsis mutants would grow better and more rapidly than those grown under short-day photoperiod conditions.
Comments for Results and Discussion
- Kindly improve on the discussion. What is the significance of the results of the work?
Response: Thank you for your suggestions. And we have carefully improved our discussion to strengthen the significance of the work.
2、Please give more authors’ opinion before including a literature reference as support, what does the authors think of the results and why does the results act so? After mentioning these then only place the citations to support the authors’ claims.
Response: Thank you for your suggestions. We have carefully improved our discussion in the revised draft.
3、 Please link the results together, the authors did not describe what does the results mean and how are the results interrelated.
Response: Thank you for your suggestions. We have discussed how these results systematically proved the high bioactivity of 28-HBL at multiple levels on page 10, paragraph 2.
4、Please improve on figures, simple line graphs may not be considered as a complete figure for some journals.
Response: Thank you for your advice. In fact, we have improved our figures as much as possible. And we thought that the results could be clearly shown by the figures in our manuscript.
5、 Please revise the discussion section, the current discussion did not link the results with the discussion, report a result then discussion on that result, what are the significance of that results to the growth of the plant? The following papers may aid in the writing of the paper: “Mesorhizobium improves chickpea growth under chromium stress and alleviates chromium contamination of soil” and “The Effects of Biofertilizers on Growth, Soil Fertility, and Nutrients Uptake of Oil Palm (Elaeis Guineensis) under Greenhouse Conditions”.
Response: Thank you for your advice. The discussion section has been updated to highlight the link between the results and the discussion. Changes and additions are in yellow.
6、 Include more author insights on each results and discussion provided, this paper lacks author insights on each result obtained by the author.
Response: Thank you for your suggestions. We have improved our presentations to highlight our insights.
Comments for Conclusions
- Conclusion section is missing, please include a conclusion section to properly wrap up the whole manuscript.
Response: Thank you for your advice and we did not prepare a conclusion section because “This section is optional”. And we thought that our conclusion had been included in the discussion section.
Comments for References:
1、Kindly revise reference format according to the author guideline.
Response: Thank you for your advice, and we have revised the reference format according to the author’s guidelines.
2、 It is suggested to cite references within 5 years of research to maintain the reliability of results obtained.
Response: Thank you for your advice, and we have added some references published recently (within 5 years) and also kept the references published in earlier years that reported original and classical protocols or gene functions.
3、 There are references found to be outdated.
Response: Thank you for your advice, and we have checked the references again and kept the classical references in the manuscript.
Round 2
Reviewer 1 Report
The changes made increase the scientific level of this article. However, I still recommend this minor revision. Figures 2,3,5,6: statistical analysis is missing. Should be completed and processed in the same way as Figure 1. Please describe the methodology of statistical analysis in Materials and Methods section.Author Response
Dear Reviewer,
Thank you again for the insightful comments that helped improve our paper “Assessment of biological activity of 28-homobrassinolide via a multi-level comparative analysis” for publication in the International Journal of Molecular Sciences. Please see below, for a point-by-point response to your kind comments.
Your helpful comments to us:
"The changes made increase the scientific level of this article. However, I still recommend this minor revision. Figures 2,3,5,6: statistical analysis is missing. Should be completed and processed in the same way as Figure 1. Please describe the methodology of statistical analysis in Materials and Methods section."
Response: Thank you again for your positive comments and valuable suggestions to improve our manuscript. We have included the “statistical analysis” of Figure 2 (Page 5, lines 211-213 and Figure 2b), Figure 3 (Page 6, lines 247-249 and Figure 3b), Figure 5 (Page 8, lines 333-335 and Figure 5b), and Figure 6 (Page 9, lines 372-373 and Figure 6b) in the revised manuscript and also described the methodology of statistical analysis in the Materials and Methods section on Page 13, lines 569-575.
Reviewer 3 Report
The manuscript is corrected and revised according to the reviewer's comments. I am now satisfied with the new version, so I would like to recommend its publication.
Minor editing of English language required
Author Response
Dear Reviewer,
Thank you again for for the insightful comments that helped improve our paper “Assessment of biological activity of 28-homobrassinolide via a multi-level comparative analysis” for publication in the International Journal of Molecular Sciences. Please see below, for a point-by-point response to the your kind comments.
Your helpful comments to us:
"Minor editing of English language required."
Response: Thank you very much for your helpful comments, and we have made careful modifications again and carefully proof-read the manuscript to minimize typographical and grammatical errors.